# Quantification of Credal Uncertainty in Machine Learning: A Critical Analysis and Empirical Comparison

**Eyke Hüllermeier**[1]       **Sebastian Destercke**[2]       **Mohammad Hossein Shaker**[1]

[1]Institute of Informatics, University of Munich (LMU), Germany
[2]UMR CNRS 7253 Heudiasyc, Sorbonne Universités, Université de Technologie de Compiegne, France

## Abstract

The representation and quantification of uncertainty has received increasing attention in machine learning in the recent past. The formalism of credal sets provides an interesting alternative in this regard, especially as it combines the representation of epistemic (lack of knowledge) and aleatoric (statistical) uncertainty in a rather natural way. In this paper, we elaborate on uncertainty measures for credal sets from the perspective of machine learning. More specifically, we provide an overview of proposals, discuss existing measures in a critical way, and also propose a new measure that is more tailored to the machine learning setting. Based on an experimental study, we conclude that theoretically well-justified measures also lead to better performance in practice. Besides, we corroborate the difficulty of the disaggregation problem, that is, of decomposing the amount of total uncertainty into aleatoric and epistemic uncertainty in a sound manner, thereby complementing theoretical findings with empirical evidence.

## 1 INTRODUCTION

In the literature, two inherently different sources of uncertainty are commonly distinguished, referred to as *aleatoric* and *epistemic* [Hora, 1996]. While the former refers to variability due to inherently random effects, the latter is uncertainty caused by a lack of knowledge and hence relates to the epistemic state of an agent. Thus, epistemic uncertainty can in principle be reduced on the basis of additional information, while aleatoric uncertainty is non-reducible.

Related to the *representation* of different types of uncertainty (in terms of a probability distribution, a set of alternatives, or a more general representation) is the problem of *uncertainty quantification*, i.e., of quantifying the amount of uncertainty associated with a representation in terms of a single number. Uncertainty quantification has been addressed by many scholars and from various perspectives. As a quantification induces a complete order relation on representations, it allows for sorting them from the least to the most uncertain, thereby supporting reasoning, inference, and learning principles such as "least commitment", "minimum specificity", or "maximum uncertainty reduction". However, many such complete orderings or precise quantifications are possible, especially when seeing them as refinements of a partial order between representations induced by more qualitative or robust notions, such as set-inclusion [Klein et al., 2016] or cumulative dominance [Dubois and Hüllermeier, 2007]. Then, a standard way to choose a suitable measure, critically discussed in this paper, is to characterise it through sets of axioms [Bronevich and Klir, 2010].

The distinction between different types of uncertainty and their quantification has also been adopted in the recent machine learning (ML) literature [Senge et al., 2014, Kendall and Gal, 2017]. This is motivated by safety-critical applications, where a proper representation of uncertainty is important, along with the observation that learning algorithms such as deep neural networks in general seem to exhibit limited awareness of their own competence [Papernot and McDaniel, 2018, Sato et al., 2018]. In order to improve the uncertainty-awareness of learning algorithms, various methods have been proposed and various ways of quantifying the uncertainty in predictions have been proposed [Hüllermeier and Waegeman, 2021].

In this paper, we focus on the connection between ML and *credal sets*, i.e., (convex) sets of probability distributions, and more specifically on the related question of *uncertainty quantification*. The usefulness of credal sets for capturing uncertainty in ML has been advocated by various scholars [De Bock et al., 2014, Corani et al., 2012], most recently also in connection with the distinction between aleatoric and epistemic uncertainty [Sensoy et al., 2018]. Uncertainty quantification in credal ML can be examined from different perspectives and with different goals in mind:

*Accepted for the 38th Conference on Uncertainty in Artificial Intelligence (UAI 2022).*

- From the perspective of ML, the most important question concerns suitable uncertainty measures: How to quantify the uncertainty contained in a "credal prediction"? In the literature on credal sets (and related uncertainty formalisms), a large repertoire of such measures has been proposed, but their usefulness in the context of ML is not always very clear.

- From the perspective of credal sets and uncertainty quantification, the usefulness and performance of different measures is arguably of interest, too: In our view, ML may provide interesting *empirical evidence* in favor or against different measures, thereby complementing a more theoretical line of work. In fact, existing measures are mostly of generic nature and have been proposed on a purely axiomatic basis.

Our investigations and the empirical study presented in this paper led to the following observations and insights:

- Measures that exhibit desirable mathematical properties show better performance than measures that are theoretically (axiomatically) less well supported — in this sense, practice confirms theory.

- The theoretical difficulty of the *disaggregation problem*, that is, of decomposing the amount of total uncertainty into aleatoric and epistemic uncertainty, is not without reason. On the contrary, there are good reasons to question the usefulness of a disaggregation of that kind.

- Properties and axioms should be tailored to the purpose of a measure and the application at hand, as not all axioms are equally important or useful for all applications.

## 2 UNCERTAINTY IN ML

We consider a standard setting of supervised learning, in which a learner is given access to a set of (i.i.d.) training data $\mathcal{D} := \{(\boldsymbol{x}_i, y_i)\}_{i=1}^N \subset \mathcal{X} \times \mathcal{Y}$, where $\mathcal{X}$ is an instance space and $\mathcal{Y}$ the set of outcomes that can be associated with an instance. In particular, we focus on the classification scenario, where $\mathcal{Y} = \{y_1, \ldots, y_K\}$ consists of a finite set of class labels, with binary classification ($\mathcal{Y} = \{0, 1\}$) as an important special case. We denote by $\Delta_K = \mathbb{P}(\mathcal{Y})$ the set of all probability measures on $\mathcal{Y}$.

Suppose a *hypothesis space* $\mathcal{H}$ to be given, where a hypothesis $h \in \mathcal{H}$ is a mapping $\mathcal{X} \longrightarrow \Delta_K$. Thus, a hypothesis maps instances $\boldsymbol{x} \in \mathcal{X}$ to probability distributions on outcomes. The goal of the learner is to induce a hypothesis $h^* \in \mathcal{H}$ with low risk (expected loss)

$$R(h) := \int_{\mathcal{X} \times \mathcal{Y}} \ell(h(\boldsymbol{x}), y) \, d P(\boldsymbol{x}, y) \ , \tag{1}$$

where $P$ is the (unknown) data-generating process (a probability measure on $\mathcal{X} \times \mathcal{Y}$), and $\ell : \Delta_K \times \mathcal{Y} \longrightarrow \mathbb{R}$ a loss function. The choice of a hypothesis is commonly guided by the empirical risk $R_{emp}(h)$, i.e., the performance of a hypothesis on the training data. However, since $R_{emp}(h)$ is only an estimation of the true risk $R(h)$, the empirical risk minimizer $\hat{h} := \mathrm{argmin}_{h \in \mathcal{H}} R_{emp}(h)$ (or any other predictor) favored by the learner will normally not coincide with the true risk minimizer (Bayes predictor)

$$h^* := \underset{h \in \mathcal{H}}{\mathrm{argmin}} \, R(h) \,. \tag{2}$$

Correspondingly, there remains uncertainty regarding $h^*$ as well as the approximation quality of $\hat{h}$ (in the sense of its proximity to $h^*$) and its true risk $R(\hat{h})$.

Eventually, one is often interested in the *predictive uncertainty*, i.e., the uncertainty related to the prediction $\hat{y}_q$ for a concrete query instance $\boldsymbol{x}_q \in \mathcal{X}$. Assuming a non-deterministic dependency, part of this uncertainty is irreducible and hence of aleatoric nature: Even knowing the ground-truth conditional probability, we get

$$p(y \mid \boldsymbol{x}_q) = \frac{p(\boldsymbol{x}_q, y)}{p(\boldsymbol{x}_q)} \tag{3}$$

on $\mathcal{Y}$, and the outcome $y_q$ cannot be predicted with certainty.

In addition, a prediction $\hat{h}(\boldsymbol{x}_q)$ also involves epistemic uncertainty, because $\hat{h}(\boldsymbol{x}_q)$ is only an estimation of the distribution (3): $\hat{h}(\boldsymbol{x}_q) \approx h^*(\boldsymbol{x}_q) \approx p(y_q \mid \boldsymbol{x}_q)$. Indeed, the Bayes predictor (2), restricted by the hypothesis space $\mathcal{H}$, may not necessarily coincide with the pointwise Bayes predictor (i.e., $h^*(\boldsymbol{x}_q) \neq p(\cdot \mid \boldsymbol{x}_q)$). Moreover, the hypothesis $\hat{h}$ produced by the learning algorithm is only an estimate of $h^*$, and the quality of this estimate strongly depends on the quality and the amount of training data.

Recall that a probabilistic predictor $h \in \mathcal{H}$ is a mapping $\mathcal{X} \longrightarrow \Delta_K$. A predictor of that kind captures aleatoric but no epistemic uncertainty: By predicting a precise probability distribution $\hat{p}(\cdot \mid \boldsymbol{x}_q)$ for any query $\boldsymbol{x}_q$, it accounts for the non-determinism of the sought dependence but pretends precise knowledge about this dependence. In order to account for epistemic uncertainty, an "uncertainty-aware" predictor of the form

$$h : \mathcal{X} \longrightarrow [\![\Delta_K]\!]$$

is needed, where $[\![\Delta_K]\!]$ is a suitable second-order formalism that allows for representing uncertainty about uncertainty. An obvious example is second-order probabilities as used in Bayesian learning.

An interesting alternative is the concept of credal sets, that is, (convex) sets of probability distributions. In fact, a key motivation of the credal approach, compared to Bayesian inference, is to model a lack of knowledge in a more adequate way, essentially arguing that epistemic uncertainty is better captured by sets than distributions. In our setting, this means that a hypothesis $h$ would no longer be assumed to provide

probabilistic predictions $h(\boldsymbol{x}_q) \in \Delta_K$, but generalized predictions in the form of a credal sets $h(\boldsymbol{x}_q) = Q \subseteq \Delta_K$. In particular, total ignorance about the class would correspond to $Q = \Delta_K$ (and not to the uniform probability). Similarly, a set $A \subseteq \mathcal{Y}$, can simply be represented by $Q = \mathbb{P}(A) = \{P \in \mathbb{P}(\mathcal{Y}) \mid P(A) = 1\}$.

Our interest in this paper is less how to learn a predictor of this kind — see Shaker and Hüllermeier [2020] for a recent proposal using ensemble learning and Augustin et al. [2014, Ch. 10] for a review. Instead, we are interested in the problem of uncertainty quantification: How to quantify the uncertainty associated with a prediction $h(\boldsymbol{x}_q) = Q$? As will be seen in the next section, various uncertainty measures that could principally be used for that purpose have been proposed in the literature on credal sets (and related formalisms).

# 3 QUANTIFYING UNCERTAINTY OF CREDAL SET REPRESENTATIONS

As already mentioned in the introduction, uncertainty quantification is important and useful for many purposes. In ML, for example, it supports prediction with (partial) abstention or active learning [Nguyen et al., 2021]. Yet, it should also be clear that uncertainty quantification is a challenging and non-trivial problem, especially in light of the multi-faceted nature of uncertainty and its representation (cf. Section 2). In this section, we review and discuss the main previous proposals in the literature on credal sets, pointing out some of the issues we see in general and especially when applying them to ML problems. In the next section, we also suggest a new quantification, arguing that it overcomes at least some of the issues mentioned and fits well the purpose of uncertainty quantification in ML.

## 3.1 HARTLEY AND SHANNON

In the credal setting, sets and probabilities represent pure epistemic and pure aleatoric uncertainty, respectively. As the problem of quantifying uncertainty has been thoroughly addressed for both set theory and probability theory, and well-founded measures have been proposed in both cases, it seems useful to first recall them here.

The standard uncertainty measure in classical set theory, where uncertain information is simply represented in the form of subsets $A \subseteq \mathcal{Y}$ of alternatives deemed possible (while the complement $\mathcal{Y} \setminus A$ is excluded), is the Hartley measure [Hartley, 1928]:

$$H(A) = \log(|A|). \tag{4}$$

This measure can be justified axiomatically[1]. It is minimal

when the set is reduced to a singleton (precise information) and maximal when $A = \mathcal{Y}$ (complete ignorance).

Likewise, the most well-known measure of uncertainty of a single probability distribution $q \in \Delta_K$ is the (Shannon) entropy, which, in the case of discrete $\mathcal{Y}$, is given by

$$S(q) := -\sum_{y \in \mathcal{Y}} q(y) \log_2 q(y), \tag{5}$$

with $0 \log 0 = 0$ by definition. Again, this measure can be justified axiomatically, and different axiomatic systems have been proposed in the literature [Csiszár, 2008]. Similar to (4), entropy is minimal when a single element $y$ is assigned probability 1, and maximal for the uniform distribution $q_{uni} \equiv 1/|\mathcal{Y}|$, which is the prototypical representation of full aleatoric uncertainty.

## 3.2 EXTENSIONS OF STANDARD MEASURES

As credal sets and other generalized uncertainty theories such as evidence theory or possibility theory extend both sets and probabilities, the question of extending measures of uncertainty such as (4) and (5) has received quite a lot of attention. Notably, following the axiomatic approach underlying the measures of Shannon and Hartley, some authors have proposed axioms that a measure of uncertainty $U$ over credal sets should obey [Abellan and Klir, 2005, Jiroušek and Shenoy, 2018]:

A1 Non-negativity, range: $U$ is non-negative and upper-bounded by some value $r \in \mathbb{R}$, for example $r = \log(K)$, which is assumed for $Q = \Delta_K$ (the case of complete ignorance).

A2 Continuity: $U$ is a continuous functional.

A3 Monotonicity: If $Q \subseteq Q'$ for credal sets $Q, Q'$, then $U(Q) \leq U(Q')$.

A4 Probability consistency: $U$ reduces to standard Shannon entropy in the case where $Q$ reduces to a single probability distribution.

A5 Sub-additivity: For a (joint) credal set $Q$ on a product space $\mathcal{Y}' \times \mathcal{Y}''$ with marginals $Q'$ resp. $Q''$,

$$U(Q) \leq U(Q') + U(Q''). \tag{6}$$

A6 Additivity: The inequality (6) is an equality in the case where $Q'$ and $Q''$ are independent[2].

A first measure that satisfies the above requirements is the maximal entropy [Abellan and Moral, 2003]:

$$S^*(Q) := \max_{q \in Q} S(q). \tag{7}$$

---

[1] For example, see Chapter IX, pages 540–616, in the book by Rényi [Rényi, 1970].

[2] This presupposes a suitably defined notion of independence; see Couso et al. [2000] for a review of such notions.

It is an upper bound of Shannon entropy and commonly perceived as a reasonable measure of *total* uncertainty, although it has at least one obvious defect: $S^*(Q)$ is maximal as soon as $Q$ contains the uniform distribution $q_{uni}$, which implies an insensitivity toward further imprecisiation in the sense that $S^*(Q') = S^*(Q)$ for all $Q' \supsetneq Q$ — in particular, $S^*(\{q_{uni}\}) = S^*(\Delta_K)$.

Another well-founded measure is a generalization of the Hartley measure (4), defined as follows [Abellan and Moral, 2000]:

$$\mathrm{GH}(Q) := \sum_{A \subseteq \mathcal{Y}} \mathrm{m}_Q(A) \log(|A|), \qquad (8)$$

where $\mathrm{m}_Q : 2^{\mathcal{Y}} \longrightarrow [0,1]$ is the Möbius inverse of the capacity function $\nu_Q : 2^{\mathcal{Y}} \longrightarrow [0,1]$ defined by $\nu_Q(A) := \inf_{q \in Q} q(A)$ for all $A \subseteq \mathcal{Y}$. This measure obeys a number of desirable properties, which, as a whole, are not shared by any other generalization of the Hartley measure[Klir and Mariano, 1987]. Obviously, it does not satisfy A4, because $\mathrm{GH}(\{q\}) = 0$ for all $q \in \Delta_K$, i.e., GH vanishes as soon as the (ground truth) distribution is precisely known. This, however, is coherent with an epistemic interpretation of this measure, i.e., the idea of GH as a measure of the lack of knowledge about this distribution.

## 3.3 DISAGGREGATION

In the literature on generalized uncertainty measures (including credal sets), aleatoric and epistemic uncertainty is also referred sometimes to as *conflict* (randomness, discord)[3] and *non-specificity* [Yager, 1983]. While the latter appears to be adequately captured by (8), an equally well-justified measure of conflict in the form of an extension of Shannon entropy has not been found so far. Instead, all measures proposed in the literature turned out to be non-satisfactory, due to the violation of important properties [Klir, 2005]. As a possible way out, it was suggested to proceed from a measure of total or aggregate uncertainty, TU, and to assume an additive representation of the following type:

$$\mathrm{TU}(Q) = \mathrm{AU}(Q) + \mathrm{EU}(Q) \qquad (9)$$

Then, given $\mathrm{TU}(Q)$ and a generalized measure of epistemic uncertainty (non-specificity) $\mathrm{EU}(Q)$, a generalized measure of aleatoric uncertainty (conflict) can be *derived* via *disaggregation*, *viz.* in terms of the difference $\mathrm{AU}(Q) = \mathrm{TU}(Q) - \mathrm{EU}(Q)$. More specifically, taking (7) as a measure of total uncertainty and generalized Hartley as a measure of non-specificity, the following disaggregation was proposed

---

[3]This aspect should not be confused with the conflict understood as a partial inconsistency or incoherence of the uncertainty representation [Destercke and Burger, 2013], [Quaeghebeur, 2015], which is not considered here, as predictions are assumed to be consistent, e.g., to be normalised probabilities.

[Abellan et al., 2006]:

$$S^*(Q) = \underbrace{\big( S^*(Q) - \mathrm{GH}(Q) \big)}_{\mathrm{GS}(Q)} + \mathrm{GH}(Q) \qquad (10)$$

The first part on the right-hand side is also called the generalized Shannon entropy. However, unlike the other parts it is derived from, $S^*$ and GH, it does not enjoy "nice" axiomatic properties.

The idea of fixing the aggregate uncertainty as well as one of the two constituent uncertainties, and then deriving the second one by the difference between these two, can of course also be applied the other way around, namely by fixing (generalized) conflict and deriving (generalized) non-specificity. As a measure of generalized conflict, the lower Shannon entropy has been proposed:

$$S_*(Q) := \min_{q \in Q} S(q) \qquad (11)$$

Correspondingly, the following disaggregations of total uncertainty is obtained [Abellan et al., 2006]:

$$S^*(Q) = S_*(Q) + \big( S^*(Q) - S_*(Q) \big) \qquad (12)$$

In this case, non-specificity is defined in terms of the difference between upper and lower entropy. An obvious defect of (11) is its non-monotonicity: The lower entropy may decrease by increasing the credal set $Q$. Nevertheless, $S_*$ can still be considered as an appealing measure of conflict resp. aleatoric uncertainty, as it indeed corresponds to the natural measure of *irreducible* uncertainty: Given a credal set $Q$, $S_*(Q)$ is a lower bound of the (aleatoric) uncertainty that remains even when all epistemic uncertainty is removed, i.e., when $Q$ is reduced to a single distribution $q \in Q$ thanks to additional information.

## 3.4 CRITIQUE AND DISCUSSION

In this section, we discuss some issues and potential problems we see with the above uncertainty measures for credal sets, especially in light of their use in machine learning.

**Disaggregation.** As already said, a fully satisfactory representation of aggregate uncertainty in the form (9), with all three measures having nice theoretical properties, has not yet been found for the case of credal sets. While $S^*$ and GH appear to be well justified, this is not completely true for $S_*$ (which violates the property of monotonicity) and even less so for the "derived" measures. Klir argues that this is unproblematic as long as the measures of conflict and non-specificity complement each other in the sense that the properties are satisfied by the aggregate uncertainty.

Regardless of the technical difficulties, one may wonder whether a decomposition (9) is meaningful from a *semantical* perspective. First, in the case of credal sets, the uncertainty formalisms underlying epistemic and aleatoric uncertainty, *viz.* sets and probability distributions, are of very

different nature. Hence, why should one expect corresponding uncertainty measures to be sufficiently "commensurable" to allow simple addition? For example, while GH measures *imprecision* about the knowledge of $q \in \Delta_K$, Shannon entropy captures *randomness* on the level of outcomes $\mathcal{Y}$.

Also, epistemic and aleatoric uncertainty are two types of uncertainty on different levels: epistemic uncertainty is a kind of "meta-level" uncertainty that partly comprises aleatoric uncertainty on the "base level". Indeed, if an agent is epistemically uncertain, it is also uncertain about the (ground-truth) aleatoric uncertainty. Take the extreme case of complete ignorance ($Q = \Delta_K$) as an example. In this case, both the total and epistemic uncertainty should be maximal. Then, however, an additive decomposition forces aleatoric uncertainty to be 0, which is not meaningful. Actually, the aleatoric uncertainty is somehow contained in the epistemic uncertainty: because the agent does not know anything, it does not know the true aleatoric uncertainty either.

In the above example, the value of 0 should arguably not be seen as a measure of aleatoric uncertainty but rather as a *lower bound* on the (true) measure of aleatoric uncertainty. More generally, $S_*$ provides a bound of exactly that kind. From this point of view, one may indeed wonder what axiomatic properties one should demand. Even if a specific property is natural for a measure of uncertainty, it does not mean that the property must also hold for a *bound* on that measure. This is quite obvious for monotonicity (A3) in the case of $S_*$: although monotonicity is a natural property of a measure of uncertainty, it is also natural that $S_*$ increases when $Q$ decreases, because the smaller $Q$, the higher the lower bound on entropy. This supports the view of Klir, namely, to demand the axioms only for total uncertainty. But it also shows that one should be careful with the notion of conflict or aleatoric uncertainty in the decomposition (9), and better speak about a *lower bound* on conflict or *guaranteed* aleatoric uncertainty.

Going one step further, one may even abandon the disaggregation altogether, and argue that the two types of uncertainty should better be kept separate. Aleatoric uncertainty would naturally be specified in terms of the interval $[S_*, S^*]$ instead of a single number. As for epistemic uncertainty, both GH and the difference $S^* - S_*$ appear to be meaningful measures. The important point to notice, however, is that these measures refer to different things: GH measures the size of the set of candidate probabilities, and hence refers to uncertainty or imprecision about the ground-truth probability $q$, whereas $S^* - S_*$, as the size of the set of candidate entropies $[S^*, S_*]$, quantifies uncertainty about the aleatoric uncertainty $S(q)$ of $q$.

**Properties and axioms.** In addition to the idea of decomposition, one may critically question the reasonableness of the axioms themselves. In particular, looking at the large body of literature on generalized uncertainty measures (we

refer to Jiroušek and Shenoy [2018] for a quite exhaustive review in the case of belief functions, which can be seen as specific credal sets), it is noticeable that most measures are proposed without considering the need of a specific domain of application. This somehow deviates from Shannon's original agenda, who specifically developed his measure with an application to communication and information theory in mind (and not to quantify the uncertainty of a subjective probability, for example). That said, there are some notable exceptions. For instance, Jiroušek and Shenoy [2020] propose a measure with the goal to build multivariate uncertainty representations within belief function theory and propose axioms that are specific to this agenda.

In the context of ML, we are mostly interested in predictive uncertainty, i.e., the learner's uncertainty about the best prediction of the target variable $Y$ given a query instance $\boldsymbol{x}$. While axioms A1–A3 might still appear indisputable in this context, this is less the case for A4–A6, especially because additivity seems to be less relevant. Besides, even Shannon entropy (largely characterized through additivity) could be questioned as the right measure of aleatoric uncertainty. For example, consider the case of binary classification ($\mathcal{Y} = \{-1, +1\}$) and let the probability $q_\theta$ on $\mathcal{Y}$ be specified by $q(+1) = \theta$ and $q(-1) = 1 - \theta$. The Shannon entropy is more "sensitive" (has a steeper slope) toward the extremes of the unit intervals ($\theta$ close to 0 or 1) and less in the middle ($\theta$ close to $1/2$). While this might be meaningful from an information-theoretic point of view (where rare events are critical), it is arguably less in the context of prediction, where the critical region is around the middle point $1/2$ but not at the boundaries. In particular, due to the properties of entropy, the derived measure $EU(Q) = S^*(Q) - S_*(Q)$ is not shift-invariant: shifting an interval $[a, b]$ more toward the middle of the unit interval (around $1/2$) will decrease $EU(Q)$, shifting it more to the boundary will increase it. This appears counter-intuitive, because in (binary) classification it should just be the other way around: a boundary interval suggests less uncertainty about the prediction than an interval of the same size in the middle of the unit interval.

# 4 ANOTHER MEASURE

As suggested by our previous discussion, uncertainty measures should be tailored to a specific domain and the purpose they are used for. In the context of ML, instead of measuring the uncertainty about the true probability or the uncertainty about the (aleatoric) uncertainty, it would be natural to seek a measure of uncertainty about the best *prediction* to be made, which is related to the uncertainty about the outcome in the respective situation. Interestingly, a measure of that kind has been proposed in the context of active learning [Antonucci et al., 2012]. In this section, taking this measure as a point of departure, we develop a new measure of uncertainty, along with a decomposition into an aleatoric and an

epistemic part, specifically suitable for quantifying predictive uncertainty. We restrict ourselves to the simple (though practically relevant) case of binary classification, where predictions and outcomes are restricted to $\mathcal{Y} = \{-1, +1\}$, leaving a generalization to multinomial classification for future work.

This aforementioned measure is based on the notion of *dominance*, which is commonly adopted for decision making based on credal knowledge representation: We say that a class $y$ dominates another class $y'$ if $y$ is more probable than $y'$ for each distribution in the credal set, that is,

$$\gamma(y, y') := \inf_{q \in Q} \frac{q(y)}{q(y')} > 1 . \qquad (13)$$

Uncertainty is then considered as a lack of dominance, i.e., a situation is considered uncertain if there is no class that dominates all others. To this end, it makes sense to look at the maximum degree of dominance over all classes. In the case of binary classification, this is expressed by the score

$$u := \max\left(\gamma(+1, -1), \gamma(-1, +1)\right) . \qquad (14)$$

Note that this is actually a measure of certainty rather than uncertainty, as it will increase as we increase our certainty that a class is dominated by another. For interval-representations[4] specifying $Q$ by the constraint $q(+1) \in [a, b]$, this yields

$$\gamma(+1, -1) = \inf_{q \in Q} \frac{q(+1)}{q(-1)} = \inf_{q \in Q} \frac{q(+1)}{1 - q(+1)} = \frac{a}{1 - a} ,$$

$$\gamma(-1, +1) = \inf_{q \in Q} \frac{q(-1)}{q(+1)} = \inf_{q \in Q} \frac{1 - q(+1)}{q(+1)} = \frac{1 - b}{b} ,$$

so that (14) can be expressed as follows:

$$u(a, b) = \max\left(\frac{a}{1 - a}, \frac{1 - b}{b}\right) . \qquad (15)$$

The wider the interval $[a, b]$, the smaller the score (15), with the minimum being obtained for the case $[0, 1]$ of complete ignorance. This is well in agreement with the idea of epistemic uncertainty. In the limit, when $[a, b]$ reduces to a precise probability $(a = b)$, i.e., the epistemic uncertainty disappears, (15) is minimal for $a = b = 1/2$ and maximal for $a = b$ close to 0 or 1. This behavior is in agreement with the conception of aleatoric uncertainty. More generally, comparing two intervals of the same length, (15) will be smaller for the one that is closer to the middle point $1/2$. Thus, it seems that the credal uncertainty score (15) does indeed combine both epistemic and aleatoric uncertainty in a single measure.

As one disadvantage of the measure, one may note that it is not bounded. Moreover, one may prefer larger values to

---

[4]In the binary case, all credal sets $Q$ are of this form.

indicate higher and not lower uncertainty. Therefore, let us propose the transformation

$$\mathrm{TP}(a, b) := \frac{1}{1 + u(a, b)} = \min(1 - a, b) , \qquad (16)$$

where TP stands for *total* measure of *predictive* uncertainty. This measure takes values between 0 and 1, the latter indicating full uncertainty ($[a, b] = [0, 1]$) and the former no uncertainty ($a, b \to 0$ or $a, b \to 1$). Since $b - a$ is a natural measure of epistemic uncertainty, which coincides with GH in the case of binary classification, one may further define a decomposition of (16) into aleatoric and epistemic uncertainty as follows:

$$\mathrm{TP}(a, b) = \underbrace{\min(1 - a, b)}_{\text{total}}$$

$$= \underbrace{\min(a, 1 - b)}_{\text{aleatoric (AP)}} + \underbrace{(b - a)}_{\text{epistemic (EP)}} . \qquad (17)$$

What could be questioned is that aleatoric uncertainty is upper-bounded by $1/2$ in this case. In fact, full (total) uncertainty is only assumed for the interval $[0, 1]$, whereas the interval $[1/2, 1/2]$ has a total uncertainty of only $1/2$. In other words, complete ignorance is considered a state of knowledge that is more uncertain than perfect knowledge about the uniform distribution. This does make sense from a knowledge representation point of view, but arguably less from a predictive perspective (for which the uniform distribution is already a kind of worst case). On the other side, this property avoids the problem of (partial) insensitivity of measures like the upper entropy, which is maximal as soon as the uniform distribution is contained in the credal set — in this case, the size of the credal set does not matter anymore. Also, (16) sorts the credal sets according to their uncertainty in a quite natural way: $\mathrm{TP}(a, b) < 1/2$ for intervals completely on the left ($b < 1/2$) or completely on the right ($a > 1/2$) of $1/2$ and $\mathrm{TP}(a, b) > 1/2$ for intervals covering $1/2$ (i.e., $a \le 1/2 \le b$).

On the one side, one may speculate that this property is a necessary prerequisite for a meaningful decomposition: If a measure of total uncertainty is the sum of a measure of aleatoric and a measure of epistemic uncertainty, it can only be maximal if both types of uncertainty are fully present — which is clearly not the case for the interval $[1/2, 1/2]$, for which the epistemic uncertainty is 0. On the other side, it arguably remains a bit peculiar that aleatoric uncertainty ranges between 0 and $1/2$, whereas epistemic uncertainty takes values in $[0, 1]$, suggesting that the two types of uncertainty are measured on two different scales. Why should epistemic uncertainty be potentially twice as high as aleatoric uncertainty? As an intuitive explanation for this asymmetry, recall our argument that a positive epistemic uncertainty also implies a (potentially) positive aleatoric uncertainty, but not vice versa. In other words, if an agent

is epistemically uncertain, there is necessarily a potential aleatoric uncertainty involved, whereas an agent can be aleatorically uncertain without being epistemically uncertain.

Note that, when specifying the aleatoric uncertainty of a measure $q_\theta$ in terms of $\min(\theta, 1 - \theta)$, i.e., in terms of the closeness of $\theta$ to the middle-point $1/2$, then $\mathrm{AP}(a, b) = \min(a, 1 - b)$ does again establish a *lower bound* on the (true) aleatoric uncertainty, because $\theta \in [a, b]$ implies $\min(a, 1 - b) \leq \min(\theta, 1 - \theta)$.

To justify the measure (16) axiomatically, let us represent an interval $[a, b] \subseteq [0, 1]$ in the form $[\mu - \delta, \mu + \delta]$. There are two operations on such an interval that can increase (or vice versa decrease) uncertainty: shifting and widening. By shifting we mean moving the interval "closer to the middle", i.e., replacing $[\mu - \delta, \mu + \delta]$ by $[\mu' - \delta, \mu' + \delta]$ such that $|\mu' - 1/2| < |\mu - 1/2|$. By widening we mean increasing $\delta$, i.e., replacing $[\mu - \delta, \mu + \delta]$ by $[\mu - \delta', \mu + \delta']$ such that $\delta' > \delta$. The following theorem justifies (16) axiomatically; the proof is given in Section A in the appendix.

**Theorem 1**: Consider an uncertainty measure $U : \mathbb{I} \longrightarrow \mathbb{R}$, where $\mathbb{I}$ is the set of intervals in $[0, 1]$. If $U$ satisfies the following properties, then it is necessarily given by (16):

A1 Complete certainty: $U(0, 0) = U(1, 1) = 0$.

A2 Complete uncertainty: $U(0, 1) = 1$.

A3 Symmetry: $U(a, b) = U(1 - b, 1 - a)$ for all $0 \leq a \leq b \leq 1$.

A4 Isometry: Widening has always the same effect, regardless of the location of the interval, i.e., $U(a - \delta, b + \delta) - U(a, b)$ only depends on $\delta$ but neither on $a$ nor $b$.

A5 Shift-invariance: Shifting a precise interval $[a, a]$ has always the same effect in the sense that $U(a + \epsilon, a + \epsilon) - U(a, a)$ only depends on $|a - 1/2| - |a + \epsilon - 1/2|$ (to what extent the distance to $1/2$ has been reduced or increased) but not on $a$.

A6 Shifting and widening are equally important, i.e., the (maximal) effect of an $\epsilon$-shift is the same as the (maximal) effect of a $\delta$-widening if $\epsilon = \delta$.

Obviously, (16) also satisfies monotonicity, i.e., $\mathrm{TP}(a, b) \leq \mathrm{TP}(a', b')$ for $a' \leq a$ and $b \leq b'$. Compared to the more classical information-theoretic uncertainty measures discussed before, (16) has a more geometric flavor. Note that, in the dichotomous case, $b - a$ corresponds to both the generalized Hartley measure as well as the "size" of a credal set. Therefore, it has an information-theoretic as well as a geometric interpretation.

## 5 EXPERIMENTS

In the absence of ground-truth uncertainties, predicted uncertainties are often evaluated indirectly, for example by assessing their usefulness for improved prediction and decision making. Here, we conduct such an evaluation by producing *accuracy-rejection* (AR) curves in the setting of selective classification. An AR curve depicts the accuracy of a predictor as a function of the percentage of rejections [Hühn and Hüllermeier, 2009]: A learner that is allowed to select those instances on which to predict while abstaining on a certain percentage $p$ of predictions will predict on those $(1 - p)\%$ on which it feels most certain. Being able to quantify its own uncertainty well, it should improve its accuracy with increasing $p$, hence the AR curve should be monotone increasing (unlike a flat curve obtained for random abstention).

Here, we compare the AR curves for the different uncertainty measures discussed above (see the appendix for the derivation of explicit expressions in the Bernoulli case, i.e., where uncertainty about a binary outcome is represented in terms of an interval $[a, b]$ for the probability of the positive class):

**Aleatoric uncertainty**:

| | | | |
|---|---|---|---|
| Lower: | AL | $:=$ | $S_*$ |
| Upper: | AU | $:=$ | $S^*$ |
| Derived: | AD | $:=$ | $S^* - \mathrm{GH} = S^* - b + a$ |
| Predictive: | AP | $:=$ | $\min(a, 1 - b)$ |

**Epistemic uncertainty**:

| | | | |
|---|---|---|---|
| Based on Hartley: | EH | $:=$ | $\mathrm{GH} = b - a$ |
| Based on Shannon: | ES | $:=$ | $S^* - S_*$ |

**Total uncertainty**:

| | | | |
|---|---|---|---|
| Axiomatic: | TA | $:=$ | $S^*$ |
| Predictive: | TP | $:=$ | $\min(1 - a, b)$ |

Note that the upper entropy $S^*$ occurs twice, namely as an upper bound on the aleatoric uncertainty (AU) and as an axiomatically justified measure of total uncertainty (TA).

To produce predictions in the form of credal sets, we follow the ensemble-based (Random Forest) approach proposed by Shaker and Hüllermeier [2020]. More specifically, we performed experiments on 9 well-known data sets from the UCI repository[5].The data sets are randomly split into 70% for training and 30% for testing, and AR curves are produced on the latter. Each experiment is repeated and averaged over 100 runs. We create ensembles using the Random Forest Classifier from SKlearn. All of the hyper-parameters are set as the default setting, with the size of ensemble equal to 100 trees. Probabilities are estimated by (Laplace-corrected) relative frequencies in the leaf nodes of a tree.

Fig. 1 shows the AR curves for four of the data sets, while the other plots are moved to Section D in the appendix. Even without any further (numerical) analysis, the results convey a very clear picture: All measures are performing quite well, except for the derived measures AD and ES, which are significantly worse and yield AR curve that remain visibly

---

[5] http://archive.ics.uci.edu/ml/index.php

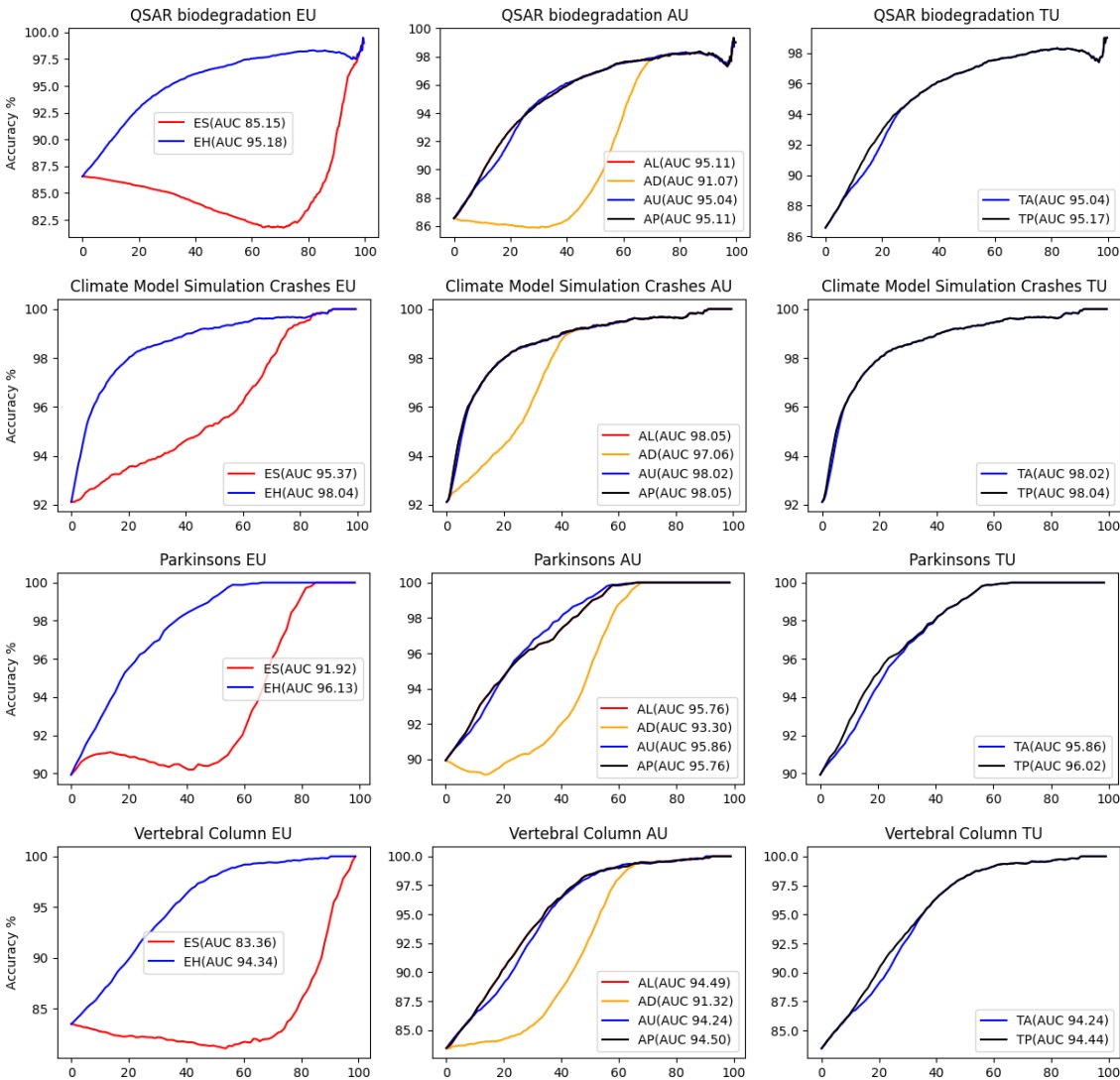

Figure 1: Accuracy-rejection curves for four data sets and different uncertainty measures (epistemic on the left, aleatoric in the middle, total on the right).

below the others. As a consequence, our newly proposed measure yields the only decomposition of total into aleatoric and epistemic uncertainty (TP = EH + AP), such that all three measures produce meaningful results — for the other decompositions, either the measure of aleatoric uncertainty fails or the measure of epistemic uncertainty.

# 6 CONCLUSION

The distinction between aleatoric and epistemic uncertainty has received increasing attention in the recent machine learning literature. In light of this, credal uncertainty representation and ML methods making use of such representations are of great interest. In this paper, assuming an ML method producing predictions in the form of credal sets, we ad-

dressed the question of *uncertainty quantification*, i.e., quantifying the amount of uncertainty contained in a prediction. In this regard, we isolated potential deficiencies of existing measures and decompositions of total into aleatoric and epistemic uncertainty. These deficiencies could be confirmed in an empirical study, in which the ML algorithm is allowed to abstain from predictions in cases where uncertainty is high. To overcome these problems, we proposed a new measure as well as its decomposition into total, aleatoric, and epistemic uncertainty. This measure can be justified theoretically and shows strong empirical performance.

As a next step, we plan to extend our measure from the dichotomous to the polychotomous case, so as to make it amenable to multi-class classification, and to apply it in related machine learning settings.

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

## A  PROOF OF THEOREM 1

Let us represent an interval $[a, b] \subseteq [0, 1]$ in the form $[\mu - \delta, \mu + \delta]$, where $0 \leq \mu \leq 1$ is the midpoint and $0 \leq \delta \leq \min(\mu, 1 - \mu)$ the width of the interval. There are two operations on such an interval that can increase (or vice versa decrease) uncertainty: shifting and widening. By shifting we mean moving the interval "closer to the middle", i.e., replacing $[\mu - \delta, \mu + \delta]$ by $[\mu' - \delta, \mu' + \delta]$ such that $|\mu' - 1/2| < |\mu - 1/2|$. By widening we mean increasing $\delta$, i.e., replacing $[\mu - \delta, \mu + \delta]$ by $[\mu - \delta', \mu + \delta']$ such that $\delta' > \delta$.

To prove Theorem 1, note that every interval $[a, b]$ is equivalently expressed in terms of $[\mu - \delta, \mu + \delta]$, where $\mu = (a + b)/2$ and $\delta = (b - a)/2$. In the following, we use the uncertainty measure with both types of arguments, $(a, b)$ and $(\mu, \delta)$. To avoid confusion, we write $U$ in the former and $U'$ in the latter case, i.e., $U'(\mu, \delta) = U(\mu - \delta, \mu + \delta)$ and vice versa $U(a, b) = U((a + b)/2, (b - a)/2)$.

Due the symmetry property A3, we can restrict our consideration to the subset $\mathbb{I}'$ of those intervals $[a, b]$ in $\mathbb{I}$ for which $1 - a \geq b$, and for which (16) takes the value $b$. Indeed, suppose $U$ is defined on $\mathbb{I}'$. Consider $[a, b] \notin \mathbb{I}$, i.e., $1 - a < b$, and let $a' = 1 - b$, $b' = 1 - a$. Then $1 - a' = b > 1 - a = b'$, hence $[a', b'] \in \mathbb{I}$ and $U(a, b) = U(a', b')$.

Note that $1 - a \geq b$ implies $\mu = (a + b)/2 \leq 1/2$. Thus, we need to consider $U$ on those intervals $[\mu - \delta, \mu + \delta]$ for which $0 \leq \mu \leq 1/2$ and $0 \leq \delta \leq \mu$.

A4 implies that, for any $\delta' > 0$,

$$\frac{U'(\mu, \delta + \delta') - U'(\mu, \delta)}{\delta'}$$

is a constant, and hence (by letting $\delta \to 0$)

$$\frac{\partial U'(\mu, \delta)}{\partial \delta} = c'$$

for a constant $c' \geq 0$. Similarly, A5 implies that

$$\frac{\partial U'(\mu, 0)}{\partial \mu} = c$$

for a constant $c \geq 0$. As a consequence, together with $U(0, 0) = 0$ according to A1, $U'$ is of the form

$$U'(\mu, \delta) = c \cdot \mu + c' \cdot \delta \,.$$

Moreover, since $U(0, 1) = U'(1/2, 1/2) = 1$ according to A2, $c/2 + c'/2 = 1$, and hence $c + c' = 2$. Finally, A6 implies $c = c'$, so that $U'(\mu, \delta) = \mu + \delta$, or equivalently,

$$U(a, b) = \frac{a + b}{2} + \frac{b - a}{2} = b = \min(1 - a, b) \,.$$

Likewise, again exploiting symmetry, we find that $U(a, b) = 1 - a$ in the case $1 - a < b$ (for which $\mu > 1/2$). Therefore, $U(a, b) = \min(1 - a, b)$ for all $[a, b] \in \mathbb{I}$.

## B  ILLUSTRATION: COIN TOSSING

As an illustration, let us consider (biased) coin tossing as a simple example: Given a sequence of outcomes so far, the problem is to predict the outcome of the next toss. Essentially, this is a problem of learning the parameter $\theta$ of a Bernoulli distribution, which corresponds to the bias of the coin. Although it may look like a toy example, this problem is actually quite relevant for ML, namely for estimating the probability $\theta$ of the positive class in binary classification, assuming this probability to be constant in a certain region of the instance space (like in nearest neighbor classification or decision tree learning).

Note that $\theta \in [0, 1]$ specifies the probability distribution $(\theta, 1 - \theta) \in \mathbb{P}(\{0, 1\})$, so that a credal set (a subset of $\mathbb{P}(\{0, 1\})$) can simply be represented by an interval $C \subseteq [0, 1]$. Credal inference can be done with the imprecise Dirichlet model, which, after $n$ coin tosses, leads to a credal set of the form

$$C = \left[ \frac{a}{n + s}, \frac{a + s}{n + s} \right] \,, \tag{18}$$

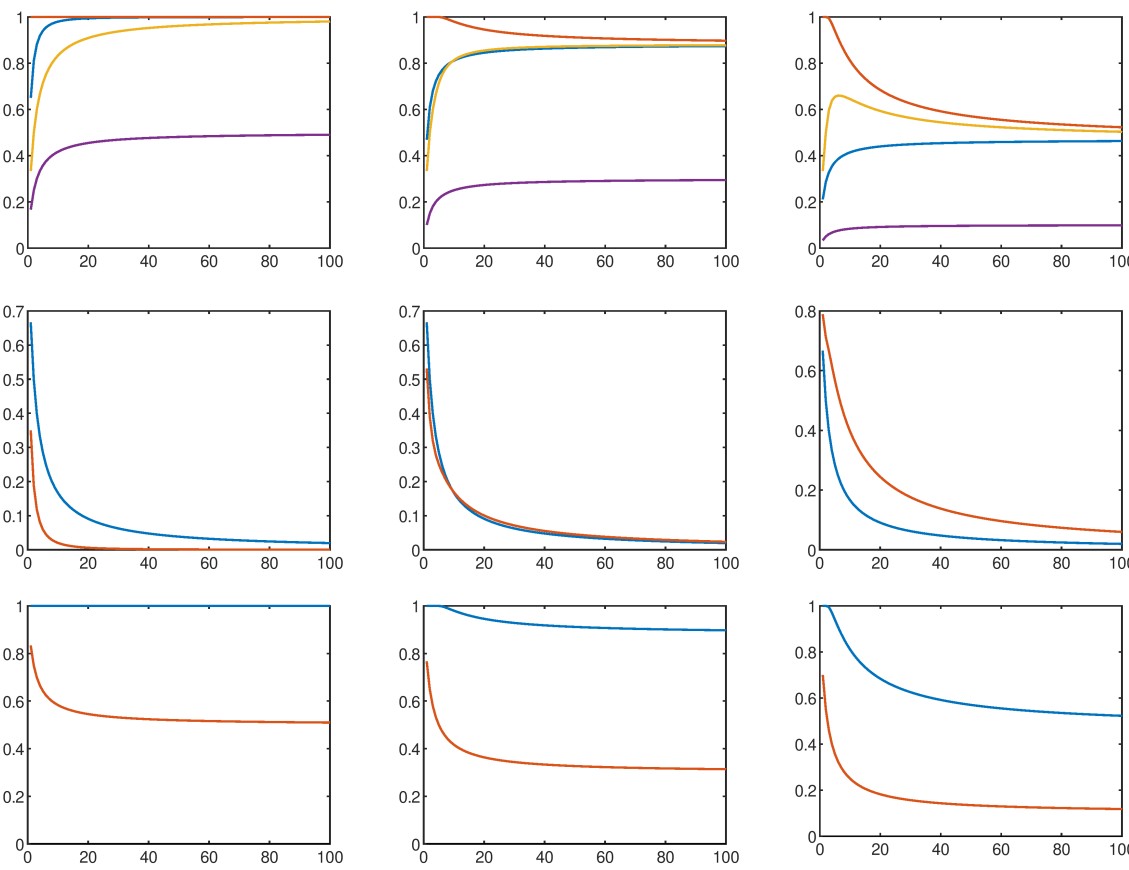

Figure 2: Upper panel: Aleatoric uncertainties AU (red), AL (blue), AD (yellow), AP (purple) for $\theta = 0.5$ (left), $\theta = 0.7$ (middle), $\theta = 0.9$ (right). Middle panel: Epistemic uncertainties EH (red), ES (blue) for $\theta = 0.5$ (left), $\theta = 0.7$ (middle), $\theta = 0.9$ (right). Lower panel: Total uncertainties TA (blue), TP (red) for $\theta = 0.5$ (left), $\theta = 0.7$ (middle), $\theta = 0.9$ (right).

where $a$ is the number of times the positive class occurred (coin landed heads up) and $s > 0$ is a parameter of the imprecise Dirichlet model (we take $s = 2$, which is often recommended in the literature).

The (expected) curves for the different uncertainty measures, i.e., the value of the respective measure ($y$-axis) as a function of the number of trials ($x$-axis), are shown in Figures 2. Basically, the results confirm our expectation, though a few notable observations can be made.

As for aleatoric uncertainty, upper entropy is a monotone decreasing and lower entropy a monotone increasing function, forming an interval for the "true" entropy. Interestingly, the derived measure AD does not necessarily behave monotonically. For example, in the case $\theta = 0.9$, it first increases and then decreases, although both epistemic and total uncertainty are monotonically decreasing. This may appear somewhat questionable, just like the (semantic) interpretation of the measure itself. The measure AP takes smaller values overall and, as discussed above, cannot exceed the value $1/2$. As can be seen in the case $\theta = 0.5$, this value is assumed for the interval $[0.5, 0.5]$.

As for epistemic uncertainty, the curves are monotonically decreasing, which is clearly expected. Remarkably, however, according to the derived measure ES, the epistemic uncertainty is overall higher for $\theta = 0.9$ than for $\theta = 0.7$, which in turn is higher than the uncertainty for $\theta = 0.5$. Again, this appears to be somewhat counter-intuitive. This problem is obviously related to the lack of shift-invariance of upper entropy (differences in entropy are smaller around $1/2$ and bigger in the boundary regions).

As shown by our discussion so far, every attempt at combining the classical measures of aleatoric (conflict) and epistemic uncertainty (non-specificity) for credal sets turns out to be problematic. This is true for both AD as a derived measure of aleatoric uncertainty as well as for ES as derived measure of epistemic uncertainty. We consider this as an affirmation of our conjecture that the two types of uncertainty are of different nature and hence not fully compatible — implying that a decomposition of total into aleatoric and epistemic uncertainty may simply not work. We are left with GH as a natural measure of epistemic uncertainty and upper entropy as a meaningful measure of total uncertainty.

Apparently, the decomposition of the measure TP into AP and EP is more meaningful.

## C   SUMMARY OF MEASURES

In the following, we summarize the measures that have been proposed. We also provide expressions for the Bernoulli case, i.e., where uncertainty about a binary outcome is represented in terms of an interval $[a, b]$ for the probability of the positive class. In this case, $S^*$ corresponds to the entropy (5) for the distribution $q$ such that

$$q(+1) = \begin{cases} b & \text{if } b < 1/2 \\ a & \text{if } a > 1/2 \\ 1/2 & \text{otherwise} \end{cases},$$

i.e.,

$$S^* = \begin{cases} S(1/2, 1/2) = \log(2) & \text{if } 1/2 \in [a, b] \\ \max(S(a, 1-a), S(b, 1-b)) & \text{otherwise} \end{cases}$$

Likewise, $S_*$ corresponds to (5) for the distribution $q$ such that

$$q(+1) = \begin{cases} a & \text{if } b < 1/2 \\ b & \text{if } a > 1/2 \\ a & \text{if } 1/2 \in [a, b], 1/2 - a > b - 1/2 \\ b & \text{if } 1/2 \in [a, b], 1/2 - a \leq b - 1/2 \end{cases},$$

i.e., $S_* = \min(S(a, 1-a), S(b, 1-b))$.

**Aleatoric uncertainty**:

| | | | |
|---|---|---|---|
| Lower: | AL | $=$ | $S_*$ |
| Upper: | AU | $=$ | $S^*$ |
| Derived: | AD | $=$ | $S^* - \text{GH} = S^* - b + a$ |
| Predictive: | AP | $=$ | $\min(a, 1-b)$ |

**Epistemic uncertainty**:

Based on Hartley: $\quad$ EH $\quad = \quad$ GH $= b - a$

Based on Shannon: $\quad$ ES $\quad = \quad S^* - S_*$

**Total uncertainty**:

Axiomatic: $\quad$ TA $\quad = \quad S^*$

Predictive: $\quad$ TP $\quad = \quad \min(1 - a, b)$

Note that the upper entropy $S^*$ occurs twice, namely as an upper bound on the aleatoric uncertainty (AU) and as an axiomatically justified measure of total uncertainty (TA).

As an aside, let us note that one may also think of the sum

$$S_* + \text{GH} = S_* + (b - a) \tag{19}$$

as a measure of total uncertainty. Indeed, given $S_*$ and GH as well-justified measures of aleatoric and epistemic uncertainty, respectively, (19) appears quite natural. Besides, this measure could also be motivated by the decomposition (17). In fact, both expressions proceed from the generalized Hartley measure $b - a$ as a natural measure of epistemic uncertainty, to which they add an "optimistic" measure of aleatoric uncertainty: As explained above, just like the lower entropy $S_*$, the term $\min(a, 1 - b)$ in (17) specifies a lower bound, as it corresponds to the "best" case in the sense of the most extreme among all conceivable probabilities. However, one can easily verify that (19), unlike (17), is not monotone, which is very undesirable for a measure of total uncertainty.

# D   ACCURACY-REJECTION PLOTS

As motioned in Section 5, the evaluation of the uncertainty measures are done indirectly with the use of accuracy-rejection curves. The idea is to sort all of test instances based on their uncertainty and start rejecting the prediction from the most uncertain to the least uncertain, and then calculate the accuracy of the model on the test data-points that remain. Ideally, we expect the highly certain predictions to be correct most of the times. Fig. 3 shows the AR curves for the five UCI data sets that are not shown in the main paper.

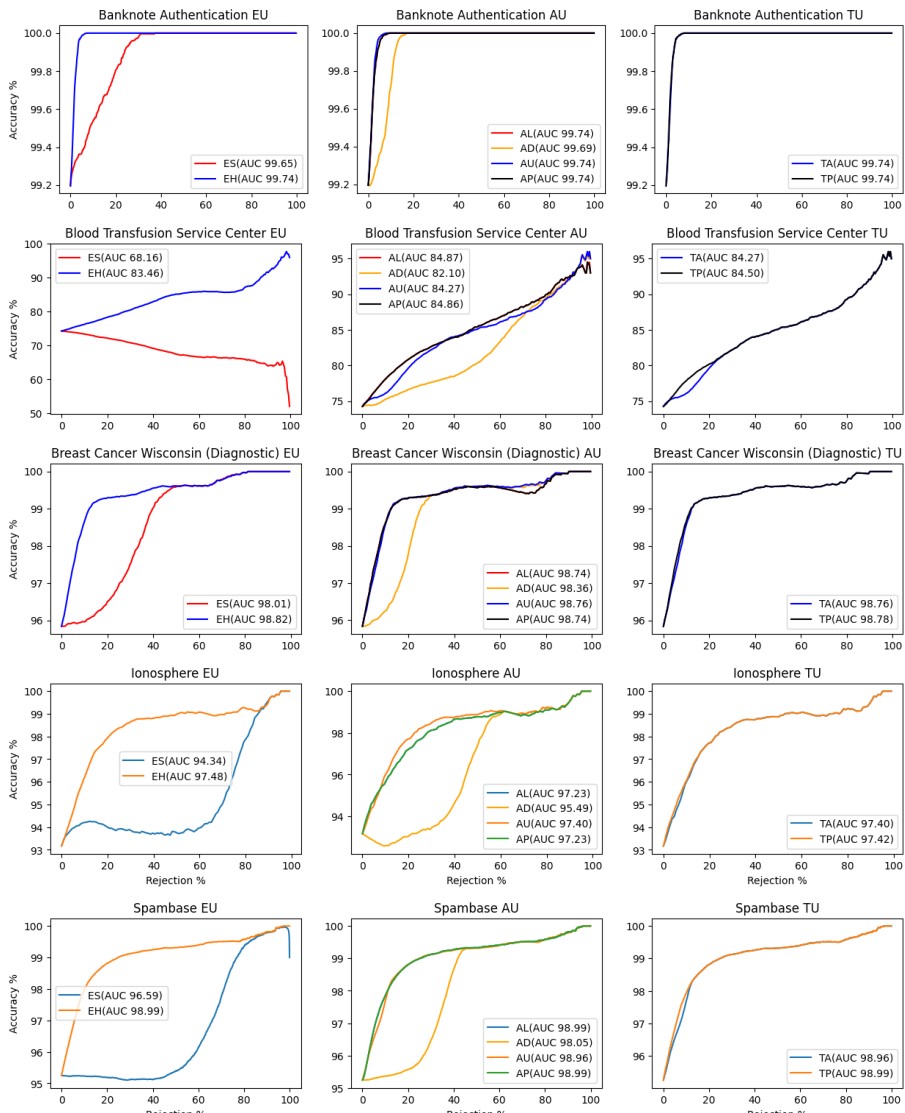

Figure 3: Accuracy-rejection curves for all the uncertainty measures summarized in Section C, separated into epistemic uncertainty on the left, aleatoric uncertainty in the middle, and total uncertainty on the right.