# OpenReview forum: "Quantification of Credal Uncertainty in Machine Learning: A Critical Analysis and Empirical Comparison"
_auai.org/UAI/2022/Conference — UAI 2022 Oral_

### Official Review · Reviewer_QSeh · 2022-04-06

**Q2(1) Originality/Novelty:** 2
**Q2(2) Significance/Impact:** 3
**Q2(3) Correctness/Technical Quality:** 3
**Q2(6) Clarity Of Writing:** 4
**Q6 Overall Score:** 7
**Q8 Confidence In Your Score:** 4

**Q1 Summary And Contributions:**

The authors present a critique to common uncertainty quantification measures for models based on sets of probability distributions. After reviewing the main proposals and discussing their pitfalls and limitations, the authors propose a new measure based on predictive uncertainty for binary classification tasks. They show an axiomatic characterization of the metric; empirical results corroborate the adequacy of the proposed metric and discredit some previous approaches based on decomposition.

**Q2 Assessment Of The Paper:**

More detailed information regarding each of these aspects is given below:

**Q2(4) Quality Of Experiments (Optional):**

3: Good: The experimental evaluation is adequate, and the results convincingly support the main claims.

**Q2(5) Reproducibility:**

3: Good: Key resources (e.g., proofs, code, data) are available and key details (e.g., proofs, experimental setup) are sufficiently well-described for competent researchers to confidently reproduce the main results.

**Q3 Main Strengths:**

- Clear and quite enjoyable reading
- Uncertainty quantification and disentanglement is an important and often overlooked topic in ML and AI.
- Thorough discussion on existing approaches for quantification of set-valued probabilistic models
- Axiomatic characterization of a new measure
- Empirical results support claims

**Q4 Main Weakness:**

- Impact and relevance of the contribution (the new measure) is not properly stated
- New metric is limited to binary classification tasks; extending to more complex tasks such as multilabel/structured classification and  regression seems non-trivial (and no hint is given in that direction by authors)
- Empirical results are somewhat weak, as they consider a small collection of datasets, whose selection is not properly motivated.
- Empirical results do not contribute to the main theme of the paper, namely, of uncertainty quantification decomposition into aleatoric and epistemic parts

**Q5 Detailed Comments To The Authors:**

The text is very enjoyable and accessible, and I am generally very positive about this work. Some points could be refined to make the discussion more solid and relevant to a larger audience.

Given its motivation, the text falls short of motivating the use of credal sets to a larger ML/AI audience. If space is of matter here, one can probably find space by compressing some trivial parts, such as inlining equations (2, 3, 5, etc).

Another criticism is the relation with Machine Learning. Once one abstracts the objective of fitting a predictive distribution p(y|x) from data, there is not much left that concerns machine learning in this work. In particular, data is absent from nearly all of the discussion. While one could advocate the dismissal of the data as a feature of the framework, one cannot but consider the disconnection with other evaluation measures used in machine learning/statistics (e.g., empirical risk, scoring, calibration). For example, instead of considering extreme elements of a credal set to characterize the aleatoric part of uncertainty, one could entertain data-driven elements such as maximum likelihood estimates. This is actually a typical case when evaluating credal models that are obtained as relaxation of precise statistical models: one compares accuracy of the precise (MLE) model when the credal model issues determinate/indeterminate predictions (one can read that as, when forced to make a prediction, use the data at hand). Putting the data at side, one could also envisage other non-extreme forms of selecting a model to measure aleatoric uncertainty such as the pignistic transformation (the center of mass of the set), or an averaged entropy. Such metrics are not subject to most of the criticisms of the extreme cases; thus, the fact that "difference metrics" perform poorly could be due to a poor choice of the aleatoric uncertainty metric, which as mentioned in the text, is only a lower/upper bound for the true aleatory uncertainty.

The experiments succeed in dismissing the proposed metrics obtained by difference from additive total uncertainty metrics; however, they do not show any advantage in actually decomposing total uncertainty into aleatoric and epistemic parts. I reckon this is due to the nature of the analysis (reject-accuracy) curves and task, which do not distinguish mistakes that originate from lack of knowledge and randomness (or to irreducible/reducible uncertainty). A typical way to distinguish both concepts is by considering performance as the sample size increases. This reintroduces the role of data in assessing uncertainty quantification; we expect a metric of aleatoric uncertainty to remain stable as sample size increases past some threshold, while a metric for epistemic uncertainty should eventually vanish and become useless for selective classification.

The authors write that "In the context of ML, we are mostly interested in predictive uncertainty, i.e., the learner’s uncertainty about the best prediction of the target variable Y given a query instance x. While axioms A1–A3 might still appear indisputable in this context, this is less the case for A4–A6, especially because additivity seems to be less relevant." I agree with A4 being less relevant, but A5 and A6 remain relevant if we consider structured prediction tasks or multilablel classification. Since a general metric is discussed, these more general cases shouldn't be so easily dismissed.

I find axiom A4 (isometry) somewhat disputable. For two intervals of same length, one containing 1/2 and the other not, it seems more desirable that the former has higher total uncertainty (as it will have higher aleatoric uncertainthy and identical epistemic uncertainty); hence, it is not unreasonable to assign higher uncertainty to an interval that grows wider to include less informative distributions than one that includes only less informative distributions.

Shaker and Hüllermeier [2020] do not explicitly consider sets of predictive distributions, instead directly obtain aleatoric and epistemic quantifications from the ensemble (random forest). Please describe in more details how the credal set is obtained, or how each metric is computed (I suppose each tree in the ensemble is used to produce a probabilistic prediction, and the set of probabilities defines the values a and b). Please justify the selection of datasets. Are they representative in any sense (small sample size, interesting domains, difficult to classify, etc)?

One disadvantage of learning a credal set from an ensemble (which is why I believe is being done with the use of random forests) is that there is no guarantees that variability in the predictions actually correspond to epistemic uncertainty. That is, random forests are designed to maintain some variability even under abundant data. Thus, one of the reasons that the empirical results do not seem to differentiate aleatoric and epistemic uncertainty is the choice of credal model. It could be interesting to evaluate the same metrics under more principled credal models, such as the Tree-Augmented Naive Credal Classifier or Credal Sum-Product Networks.

**Q7 Justification For Your Score:**

The paper is quite enjoyable to read; the authors did an excellent work in connecting different topics in the literature, and found a niche topic (decomposition of uncertainty quantification in credal models) that is relevant and little explored. Besides the interesting literature review, the work contributes a new metric with an axiomatic characterization. Empirical results are also welcome. The main weaknesses is the limited scope of the metric to binary classification tasks.

**Q9 Complying With Reviewing Instructions:**

1: Yes.

---

### Official Review · Reviewer_iZyd · 2022-04-07

**Q2(1) Originality/Novelty:** 3
**Q2(2) Significance/Impact:** 2
**Q2(3) Correctness/Technical Quality:** 3
**Q2(6) Clarity Of Writing:** 4
**Q6 Overall Score:** 6
**Q8 Confidence In Your Score:** 3

**Q1 Summary And Contributions:**

The paper investigates uncertainty measures of credal sets, i.e., convex sets of probability distributions. It contains a critical discussion on several existing measures and proposes a new measure specially for binary classifications. It is shown that the proposed measure is the sum of aleatory and epistemic uncertainties. A characterization theorem with respect to some postulates is proved and experimental comparison with other measures is given.

**Q2 Assessment Of The Paper:**

More detailed information regarding each of these aspects is given below:

**Q2(4) Quality Of Experiments (Optional):**

3: Good: The experimental evaluation is adequate, and the results convincingly support the main claims.

**Q2(5) Reproducibility:**

3: Good: Key resources (e.g., proofs, code, data) are available and key details (e.g., proofs, experimental setup) are sufficiently well-described for competent researchers to confidently reproduce the main results.

**Q3 Main Strengths:**

The exposition is clear and the proposed measure is well-justified theoretically and experimentally. The critical discussion is insightful.

**Q4 Main Weakness:**

The restriction to the binary classification setting seemingly limits the generality and impact of the proposed measure. In addition, there is a theoretical concern on whether it is appropriate to take total uncertainty as the summation of aleatory and epistemic uncertainty.

**Q5 Detailed Comments To The Authors:**

While the experimental comparison shows that the proposed measure behaves similarly with several existing ones in terns of AR curves, most of them (such as AL, AU, EH, and TA) are defined for general credal sets, as TP or AP are only defined for credal sets of probability distributions on binary domains. This seemingly restricts the potential application scope of the proposed measure.  Moreover, it is emphasized that the main strength of the proposed measure is its decomposability into aleatory and epistemic uncertainty. However, as it is argued that aleatory and epistemic uncertainty are two orthogonal aspects of uncertainty, does it really make sense to add them together? More specifically, aleatory uncertainty can be precisely determined only when epistemic uncertainty vanishes. When epistemic uncertainty exists, we can only estimate a range of possible aleatory uncertainties. In the proposed measure, it seems that the total  uncertainty is derived as the summation of epistemic uncertainty and the lower bound of the range of  aleatory uncertainties.  However, the choice of the lower bound seems somewhat arbitrary. Why not upper bound, average, or something else in the range? Hence, it seems that in the calculation of the total uncertainty,  some important information in the range is lost. In this respect,  I was wondering if a more sensible representation of the total uncertainty is not just a number, but a pair of aleatory uncertainty (as an interval or a subset of numbers) and epistemic uncertainty (as a number)


**Q7 Justification For Your Score:**

The idea behind the proposed measure is novel and well-justified. Its result can stimulate meaning discussion.

**Q9 Complying With Reviewing Instructions:**

1: Yes.

---

### Official Review · Reviewer_7shs · 2022-04-11

**Q2(1) Originality/Novelty:** 3
**Q2(2) Significance/Impact:** 2
**Q2(3) Correctness/Technical Quality:** 3
**Q2(6) Clarity Of Writing:** 3
**Q6 Overall Score:** 7
**Q8 Confidence In Your Score:** 4

**Q1 Summary And Contributions:**

This is a nice paper about uncertainty quantification in machine learning. The authors focus on credal formalisms (i.e., based on sets of distributions). After a review of the existing uncertainty measures, a new measure is provided together with a justification (Th1) based on desirable properties. Experiments show clear advantages wrt other measures for experiments based on UCI datasets and a credal random forests of Shaker and Hullermeier (2020).

**Q10 Ethical Concerns (Optional):**

No concerns.

**Q2 Assessment Of The Paper:**

More detailed information regarding each of these aspects is given below:

**Q2(4) Quality Of Experiments (Optional):**

3: Good: The experimental evaluation is adequate, and the results convincingly support the main claims.

**Q2(5) Reproducibility:**

3: Good: Key resources (e.g., proofs, code, data) are available and key details (e.g., proofs, experimental setup) are sufficiently well-described for competent researchers to confidently reproduce the main results.

**Q3 Main Strengths:**

The paper is heuristically presenting a new uncertainty measure. Yet, a technical result is derived to prove a number of desirable properties for such measure. The (extensive) experiments are clearly showing superior performances in that setting (and the setup seems to be easily reproducible).

The problem of splitting aleatoric and epistemic uncertainty is gaining importance and the contribution in this paper might have an impact in the ML community. Of course the focus on credal techniques (that are not mainstream in ML) might contrast this point, but I believe that the point of the authors is very clear (and clearly explained) and it deserves publication.


**Q4 Main Weakness:**

The main weakness of the paper is the focus on credal methods, that, as said, are not among the most popular approaches to ML. As the use of such set-valued techniques is paradoxically the key to obtain a good disaggregation, I am not expecting the authors to consider "precise" approaches, but better explaining how such credal prediction can be obtained from the major approaches in probabilistic machine learning.


**Q5 Detailed Comments To The Authors:**

As said, I am overall happy with the paper and the major change I would expect in the paper is a better justification and positioning of the credal approaches wrt to contemporary machine learning. I similarly miss a deeper discussion about the motivations for disaggregation (which is somehow a crucial point for the paper). No special focus is paid to the complexity of the different measures. Some comments on that would be useful.

**Q7 Justification For Your Score:**

The paper is clearly a new important results in the field of uncertainty measures for credal models. The strong point by the authors is that the proposed methods allows to obtain SOTA disaggregation. The result is based on a non-trivial axiomatic characterisation and the experiments are convincing in advocating this point.

**Q9 Complying With Reviewing Instructions:**

1: Yes.

---

### Official Review · Reviewer_oHkD · 2022-04-13

**Q2(1) Originality/Novelty:** 4
**Q2(2) Significance/Impact:** 3
**Q2(3) Correctness/Technical Quality:** 4
**Q2(6) Clarity Of Writing:** 4
**Q6 Overall Score:** 10
**Q8 Confidence In Your Score:** 4

**Q1 Summary And Contributions:**

This paper provides a critical review and discussion of existing measures of uncertainty for credal sets and proposes a novel measure which is more tailored to the machine learning setting.

**Q2 Assessment Of The Paper:**

More detailed information regarding each of these aspects is given below:

**Q2(4) Quality Of Experiments (Optional):**

3: Good: The experimental evaluation is adequate, and the results convincingly support the main claims.

**Q2(5) Reproducibility:**

4: Excellent: Key resources (e.g., proofs, code, data) are available and key details (e.g., proof sketches, experimental setup) are comprehensively described for competent researchers to confidently and easily reproduce the main results.

**Q3 Main Strengths:**

Brilliant and deep contribution

**Q4 Main Weakness:**

I cannot see any

**Q5 Detailed Comments To The Authors:**

Excellent piece of work! I found it brilliant, deep, and convincingly argued for. Top quality.

**Q7 Justification For Your Score:**

Brilliant and deep contribution, convincingly argued for and technically flawless.

**Q9 Complying With Reviewing Instructions:**

1: Yes.

---

### Decision · Program_Chairs · 2022-05-15

**Decision:**

Accept (Oral)

**Comment:**

Meta Review: The paper reviews methods to quantify the uncertainty when we model our problem with credal sets, ie, sets of distributions. This is not a mainstream topic in ML, but it is among the most well-founded proposals to extend traditional probability. The authors carefully review a number of measures and propose a principled new one for the binary case, then evaluating it experimentally.

This is a good paper, well written and from knowledgeable authors (as it is clear from the writing). The reviewers are essentially all in favour of acceptance, and the discussion has highlighted a good understanding between reviewers and authors. Some of the reviews have particularly been deep and insightful.

Overall I believe this paper should be accepted: it is focused on a key point of generalised uncertainty representations, it is serious work and well-written, and it proposes a sensible new measure with good properties.